# What Worth the Garlic Peel

**DOI:** 10.3390/ijms23042126

**Published:** 2022-02-15

**Authors:** Jeevan R. Singiri, Bupur Swetha, Arava Ben-Natan, Gideon Grafi

**Affiliations:** French Associates Institute for Agriculture and Biotechnology of Drylands, Jacob Blaustein Institutes for Desert Research, Ben-Gurion University of the Negev, Midreshet Ben-Gurion 84990, Israel; jeevannaveen01@gmail.com (J.R.S.); bupurswethanagaraju1234@gmail.com (B.S.); aravab743@gmail.com (A.B.-N.)

**Keywords:** Asexual reproduction, bulb, Allium sativum (garlic), dead organs enclosing reproductive body, garlic peels, proteome, primary metabolites, nucleases, chitinases, proteases, microbial growth, allelochemicals

## Abstract

Plants have two types of reproduction: sexual, resulting in embryo production, and asexual, resulting in vegetative bodies commonly derived from stems and roots (e.g., bulb, tuber). Dead organs enclosing embryos (DOEEs, such as seed coat and pericarp) are emerging as central components of the dispersal unit acting to nurture the embryo and ensure its survival in the habitat. Here we wanted to investigate the properties of dead organs enclosing plant asexual reproductive bodies, focusing on the garlic (*Allium sativum*) bulb. We investigated the biochemical and biological properties of the outer peel enclosing the bulb and the inner peel enclosing the clove using various methodologies, including bioassays, proteomics, and metabolomics. The garlic peels differentially affected germination and post-germination growth, with the outer peel demonstrating a strong negative effect on seed germination of *Sinapis alba* and on post-germination growth of *Brassica juncea*. Proteome analysis showed that dead garlic peels possess 67 proteins, including chitinases and proteases, which retained their enzymatic activity. Among primary metabolites identified in garlic peels, the outer peel accumulated multiple sugars, including rhamnose, mannitol, sorbitol, and trehalose, as well as the modified amino acid 5-hydroxylysine, known as a major component of collagen, at a higher level compared to the clove and the inner peel. Growth of *Escherichia coli* and *Staphylococcus aureus* was promoted by garlic peel extracts but inhibited by clove extract. All extracts strongly inhibited spore germination of *Fusarium oxysporum f.sp. melonis*. Thus, the garlic peels not only provide physical protection to vegetative offspring but also appear to function as a refined arsenal of proteins and metabolites for enhancing growth and development, combating potential pathogens, and conferring tolerance to abiotic stresses.

## 1. Introduction

The existence of plant species in natural habitats is largely dependent on their capability to reproduce and disperse their offspring. The dispersal unit, apart from having crucial importance in dispersion, also has beneficiary uses in the protection of embryos from predation and various biotic and abiotic hazards [1,2]. In sexually reproducing plants, the embryo is enveloped by multiple dead coverings that are maternally derived, such as seed coat, pericarp, and floral bracts (lemma, palea, glumes) in Poaceae species. These dead coverings are referred to as dead organs enclosing embryos (DOEEs). It is commonly believed that during cell death, macromolecules such as DNA, RNA, and proteins are degraded and their constituents remobilized into other plant parts [3,4]. In recent years, several reports have uncovered the elaborated function of DOEEs acting as long-term storage for active proteins, including hydrolases (e.g., nucleases, chitinases, and proteases), reactive oxygen species (ROS) detoxifying enzymes, and cell wall-modifying enzymes. In addition, DOEEs store and release numerous substances upon hydration, such as phytohormones, sugars, amino acids, and substances that control microbial growth and germination of heterologous species [5,6,7,8,9,10]. Many heat shock proteins (HSPs) (e.g., HSP90, HSP70, sHSPs), which are often induced in response to multiple stresses [11,12], are also accumulated in DOEEs [8,10,13]. Numerous reports highlighted the positive effect of DOEEs on seed longevity, seed germination, and seedling performance [5,13,14,15,16,17]. Thus, DOEEs appear to be an essential component of the dispersal unit that evolves in each plant species along with its unique habitat to nurture the embryo and to ensure its success in the habitat [9].

Besides sexual reproduction, plants also reproduce asexually through vegetative parts such as stems (e.g., tuber, bulb) and roots (rhizomes, adventitious buds, root tuber) [18]. Like the sexual reproductive embryo, the asexual reproductive body is surrounded by dead, protective layers whose function has not been well studied to date. Thus, we aimed to explore the properties of dead organs enclosing the asexual reproductive body using garlic (*Allium sativum*) bulb as a subject study. Garlic cloves have been extensively used as a spice and for medicinal purposes for thousands of years [19,20]. The bulb of garlic is made of multiple cloves surrounded by a papery, transparent outer peel; each clove is covered by a thick layer of inner peel (Figure 1). Notably, the term “garlic peel” is often used in the linguistic Jewish culture to describe something or somebody that is worthless. Both the outer and the inner peels are dead at maturity and provide a defense layer that protects the reproductive cloves from potential pathogens and abiotic stresses. Indeed, extracts of garlic skin and onion peels possess antimicrobial substances [21] as well as antioxidants with strong radical scavenging activity [22]. We investigated the biochemical and biological properties of garlic peels, and the functional similarity between garlic peels and DOEEs is discussed.

## 2. Results

### 2.1. Effect of Garlic Peel Extracts on Seed Germination

The garlic bulb is enclosed by a paper-like, often transparent peel (hereafter referred to as the outer peel) and composed of multiple cloves each enclosed by the so-called garlic skin (inner peel) (Figure 1). We analyzed the effect of outer and inner peel extracts on seed germination of *Sinapis alba* and *Brassica juncea* in comparison to water. To this end, seeds of *B. juncea* were germinated on blot paper in the presence of extracts derived from the outer peel, inner peel, and water, and germination was recorded after 24 and 48 h. The results showed that after 48 h *B. juncea* seeds were fully germinated under all treatments (Figure 2A). However, post-germination growth of the root was significantly reduced under the outer peel extract and to a lesser extent under the inner peel extract (Figure 2B,C). Further assays with *S. alba* showed a very strong inhibitory effect of the outer peel extract on germination (2.5% germination after 96 h) compared to the inner peel (47.5%) and water (53.75%) (Figure 2D). However, seeds germinated on the outer peel extract were partly recovered following washing and incubation in water for 48 h, reaching 53.75% germination compared to 73.75% germination of seeds initially treated with water or inner peel extract (Figure 2E). Thus, the outer garlic peel contains substances that can inhibit germination and post-germination growth of *S. alba* and *B. juncea*, respectively.

### 2.2. Garlic Peels Function as a Protein Storage Entity

Proteins derived from outer and inner peels were subjected to proteome analysis by LC-MS/MS on LTQ-Orbitrap followed by identification and quantification by MaxQuant, using *Brachypodium distachyon* proteins from UniProt as a reference (Appendix A). This analysis revealed a total of 67 proteins in outer and inner garlic peels. The Venn diagram shows (Figure 3A) that 61 proteins were recovered from the outer peel and 43 from the inner peel. Both the outer and inner peels shared 37 proteins; 24 and six proteins were unique to the outer and inner peels, respectively. Categorization for the biological process showed that among the 55 proteins identified in this category, nine proteins were related to response to stimulus (Figure 3B), including chitinases, peroxidases, and heat shock proteins (HSPs) (Table 1). Categorization for protein class revealed eight proteins related to protein-modifying enzymes (Figure 3C), all of which were members of serine and cysteine proteases (Table 1). All proteins related to response to stimulus and to protein-modifying enzymes (17 proteins) were present in the outer peel, and only seven proteins were in the inner peel (Table 1).

### 2.3. Garlic Peel Proteins Retain Their Enzymatic Activities

The proteome data highlighted certain hydrolytic proteins in garlic peels, including chitinases and proteases, and we wanted to examine their enzymatic activity by in-gel assays. In-gel chitinase assay revealed (Figure 4A) the activity of multiple chitinases extracted from outer and inner peels as well as from the clove tissue. In-gel protease assays showed the presence of several active proteases in the outer and inner peels, but almost no activity was recovered from the clove (Figure 4B). We also uncovered nuclease activity in clove tissue and the outer peel but not in the inner peel (Figure 4C).

### 2.4. Primary Metabolites in Garlic Peels

We performed metabolite profiling of the outer and inner garlic peels and clove tissue and identified 56 primary metabolites (Appendix A). A principal component analysis (PCA) of all identified primary metabolites showed differences in the accumulation of metabolites in the different organs analyzed (Figure 5A). Most of the variance (96.8%) was demonstrated in the first principal component (PC 1) separating dead organs (outer and inner peels) from the live clove. Hierarchical clustering analysis further highlighted the differences in the accumulation of primary metabolites between the examined organs (Figure 5B). As expected, the level of most metabolites was higher in the live clove tissues than in the dead garlic peels (Figure 6). Yet, the outer garlic peel accumulated multiple sugars, including rhamnose, mannitol, sorbitol, and trehalose, as well as the modified amino acid 5-hydroxylysine, at a higher level compared to the clove and the inner peels (inset in Figure 6).

### 2.5. Effect of Garlic Peel Extracts on Microbial Growth

Garlic is well known for its potent antimicrobial activity [24,25]. We examined the effect of garlic peel extracts on the growth of *Escherichia coli* and *Staphylococcus aureus.* To this end, bacteria were grown in a flat-bottom 96-well microtiter plate in LB medium supplemented with PBS, garlic peel extracts, or garlic clove extract. Ampicillin and kanamycin were also used as antibiotic references for *E. coli* and *S. aureus,* respectively. Plates were incubated in the dark using a Synergy 4 spectrophotometer (Biotek, Winooski, VT, USA), and reads (OD_595_) were taken at 30 min intervals over a course of 12 h. As expected, the garlic clove extract strongly inhibited the growth of both *E. coli* and *S. aureus*, similar to ampicillin and kanamycin (Figure 7A,B). The outer peel extracts had a promotive effect on *E. coli* and *S. aureus* growth, whereas the inner peel extract promoted *E. coli* growth but showed no effect on *S. aureus*. In contrast with their promotive effect on bacterial growth, substances extracted from all parts of the garlic bulb, including the outer and inner peels and the clove, strongly inhibited conidial germination and growth of the pathogenic fungus *Fusarium oxysporum f.sp. melonis* (Figure 7C,D).

## 3. Discussion

“All the sages of Israel appear to me as garlic peel, except for this bald man” (Ben Azzai, Babylonian Talmud, Tractate Bekhorot, page 58), from which the phrase “as garlic peel” has been extracted and perhaps wrongly assimilated into the linguistic Jewish culture for demonstrating something or somebody that is worthless. Yet, as demonstrated here, the garlic peel is by no means worthless, but rather valuable, and has a major role in protecting and ensuring the success of the vegetative reproductive bulb in the habitat. Thus, the papery, often transparent dead garlic peel surrounding the garlic bulb appears to have a similar elaborated role as the dead coverings enclosing the embryo [9].

The outer garlic peel and, to a lesser extent, the inner peel, contain allelopathic substances that selectively inhibit seed germination and post-germination growth of *S. alba* and *B. juncea*, respectively. The capacity to selectively inhibit germination of heterologous species appears to be a general theme in seed biology that allows plants to control their immediate surroundings to ensure their establishment in the niche. This phenomenon has been described in a variety of plant species [26]. Yet, germinating seeds may excrete substances that besides inhibiting germination and growth of a given species can promote or have no effect on the growth of other species [27,28]. Permitting or promoting the growth of other species may contribute to facilitative plant–plant interaction [29].

We showed that outer and inner garlic peels possess multiple beneficial bioactive substances, including primary metabolites and proteins that retain their enzymatic activity. Among the proteins accumulated in garlic peels are proteins related to response to stress, such as chitinases, peroxidases, and HSPs, as well as protein-modifying enzymes, namely, cysteine and serine proteases, which are accumulated mostly in the outer peel. Indeed, in-gel assays showed that the outer garlic peel displayed activities of chitinases, proteases, and nucleases, whereas the inner peel was deficient in nucleases and the clove was deficient in proteases. Although the exact function of these hydrolytic enzymes in the protection and sprouting of cloves is presently unknown, they are involved in multiple cellular processes that could contribute to clove longevity, sprouting, and establishment. Accordingly, chitinases, proteases, and nucleases in the outer garlic peel may represent a primary defense layer that can act as pathogen inhibitors and confer resistance against fungal pathogens [30,31,32,33,34]. Consistent with this idea is the finding that extracts derived from garlic peels and cloves strongly inhibited spore germination of *Fusarium oxysporum f.sp. melonis* (Figure 7). Interestingly, in contrast to the cloves, which strongly inhibited the growth of Gram-negative (*E. coli*) and Gram-positive (*S. aureus*) bacteria, garlic peel extracts had a promotive effect on bacterial growth. Likewise, dead organs enclosing the embryo, such as dead pericarps of *S. alba* and *B. juncea* and the husk of *Avena sterilis*, were found to accumulate substances that significantly enhanced bacterial growth; pericarps of *Anastatica hierochuntica* possess substances that strongly inhibited microbial growth [6,8,13,35,36]. Indeed, garlic skin has been reported recently to affect bacterial communities. Accordingly, a recent report demonstrated that supplementation of garlic skin improved the growth performance of lambs by altering the ruminal bacterial composition by increasing the relative abundances of certain genera while decreasing others [37]. Garlic skin also improved the fermentation quality of high-moisture silages by increasing the abundance of Lactobacillus and decreasing the relative abundance of Clostridium [38]. Notably, although garlic peels and cloves displayed different effect on microbial growth, all parts of the bulb inhibited microconidia germination and hyphae branching of the pathogenic fungus *Fusarium oxysporum f.sp. melonis*. Thus, it appears that dead organs enclosing the embryo, or the vegetative reproductive structure, have variable effects on microbial growth, which could be related to the plant-specific habitat and the coevolution with the microbiota. We assume that microbes whose growth is facilitated by substances released from garlic peels may provide sprouting garlic with growth regulators and defense inducers [39] that increase the survival rate and plant growth and development.

Primary metabolite analysis revealed multiple metabolites accumulated in garlic peels. Interestingly, the outer garlic peel accumulates multiple sugars, including rhamnose, mannitol, sorbitol, and trehalose, as well as the modified amino acid 5-hydroxylysine, at a higher level compared to the clove and the inner peel (Figure 5). Notably, 5-hydroxylysine is well known as a unique amino acid of collagen, the most abundant structural protein in vertebrates [40]. Mannitol, a six-carbon sugar alcohol found in various plant species, may be involved in plant stress responses. It is accumulated in response to salt and osmotic stresses and confers tolerance due to its function as a compatible solute [41]. Sorbitol, another six-carbon sugar alcohol, is implicated in plant stress tolerance. It is highly up-accumulated in *B. juncea* pericarps grown under salt condition [10] as well as in *Plantago maritime* growing in coastal habitats, where it confers salt tolerance [42]; exogenous application of sorbitol mitigated salt stress damage in salt-sensitive rice seedlings [43]. The accumulation of sorbitol was also induced in response to water stress [44,45,46], and an *Arabidopsis* mutant in *SORBITOL DEHYDROGENASE* (*SDH*) gene, whose product converts sorbitol to fructose, appeared to be more tolerant to dehydration stress [47]. Thus, we presume that sorbitol accumulation in the outer peel of garlic could be made, upon hydration, available for sprouting cloves, which might improve growth and confer tolerance under salt and drought conditions.

Trehalose, a disaccharide sugar consisting of two molecules of glucose, is known to be accumulated in high levels in the leaves of resurrection plants [48,49]. It may act as an osmoprotectant [50], keeping the integrity and functionality of cells during the dry season and consequently contributing to survival in dry and semidry environments [51].

Finally, threitol, a four-carbon sugar alcohol that is accumulated in the outer peel, has been implicated as a cryoprotectant (antifreeze agent) in the Alaskan beetle *Upis ceramboides* [52].

## 4. Materials and Methods

### 4.1. Collection and Preparation of Samples

Garlic bulbs (“normal white garlic” variety, China) were purchased from a local market. The bulbs were sorted, and the outer and inner peels were removed separately, ground into a fine powder, and incubated in phosphate-buffered saline (PBS, 1:10 *w/v* ratio) at 4 °C for 12 h. Samples were centrifuged (4 °C, 11,000× *g* rpm, 6 min) and supernatants were collected and kept at −20 °C until used for various analyses.

### 4.2. Germination Assays

The effect of extracts obtained from the outer and inner peels of garlic upon germination of heterologous species *Sinapis alba* L. and *Brassica juncea* L. was performed in a Petri dish on blot paper or red sandy soil supplemented with water or with peel extracts. Germination was performed in the dark at 22 °C, inspected daily, and photographed. Each treatment was performed in 5 replicates, each containing 20 seeds.

### 4.3. Proteome Analysis

Proteomic analysis of *Allium sativum* (outer and inner peels) was performed by the proteomic services of The Smoler Protein Research Center at the Technion, Haifa, Israel. Proteins released from 10 mg of garlic peels incubated in 100 µL PBS at 4 °C for 10 h with gentle rotation were digested with trypsin, followed by separation and mass measurement via liquid chromatography with tandem mass spectrometry (LC-MS/MS) on LTQ-Orbitrap (ThermoFisher Scientific, Waltham, MA, USA; https://proteomics.net.technion.ac.il/proteomic-services/ accessed on 11 May 2021). Protein identification and quantification were done using MaxQuant, using *Brachypodium distachium* proteins from Uniport as a reference. All the identified peptides were filtered with high confidence, top rank, and mass accuracy. High-confidence peptides passed the 1% FDR threshold (FDR = false discovery rate, the estimated fraction of false positives in a list of peptides). Four replicates were performed for each examined garlic peel. A peptide was considered “present” if it occurred in at least two replicates of either the outer or inner peel.

GO categorization analysis was carried out using the PANTHER classification system against the *Brachypodium distachium* UniProt database and the following GO trees were examined: biological process and protein class.

### 4.4. In-Gel Chitinase Assay

In-gel chitinase assay was performed essentially as described by Trudel and Asselin [53]. Briefly, garlic peels were incubated in 0.1 M NaHPO_4_ (pH 6) at 4 °C for 12 h, after which the supernatant was collected and taken for separation on SDS/PAGE. Samples were first incubated in chitin sample buffer (15% sucrose, 2.5% SDS, 12.5 mM Tris-HCl pH 6.7, 0.01% bromophenol blue) for 1 h at 37 °C, and samples were run on 12% SDS/PAGE containing 0.01% glycol chitin. The gel was incubated in buffer containing 100 mM sodium acetate (pH 5.2) and 1% triton x−100 for 2 h at 37 °C followed by staining for 5 min with 0.01% calcofluor white in 500 mM Tris-HCl (pH 8.9). The gel was washed with distilled water for 1 h and visualized by UV transillumination.

### 4.5. In-Gel Protease Assay

In-gel protease assay was performed essentially as described by Solomon et al. [54]. Briefly, substances released from peels of *Allium sativum* were incubated in loading gel sample buffer for 1 h at 37 °C. Then, samples were loaded and run in 12% SDS/PAGE containing 0.12% gelatin. After running was completed, the gel was washed twice each for 45 min in buffer containing 10 mM Tris-HCl (pH 7.5) and 0.25% Triton x-100 followed by overnight incubation in 10 mM Tris-HCl (pH 7.5). The gel was then incubated in 10 mM Tris-HCl (pH 7.5) containing 10 mM CaCl_2_ and 10 mM MgCl_2_ for 30 min at 30 °C and stained by Coomassie blue for 1 h at room temperature.

### 4.6. In-Gel Nuclease Assay

Nuclease activity was performed essentially as described [55] in polyacrylamide gel containing 300 µg/mL denatured salmon sperm DNA. Briefly, proteins released from peels of *Allium sativum* were incubated with sample buffer containing 250 mM Tris HCl pH 6.8, 10 mM EDTA, 10% glycerol, and 0.025% bromophenol blue at 37 °C for 1 h followed by separation on SDS/PAGE. The gel was washed twice for 30 min each with buffer containing 10 mM Tris HCl pH 7.5 and 25% isopropanol at room temperature, followed by washing with 10 mM sodium acetate pH 5.2 twice for 15 min each. The gel was incubated in 10 mM Tris HCl pH 7.5 containing 10 mM MgCl_2_, 10 mM CaCl_2_, and 10 mM ZnSO_4_ for 2 h at 37 °C, and then stained for 60–80 min with 10 mM Tris HCl pH 7.5 containing 2 μg/mL ethidium bromide and inspected under UV light.

### 4.7. Metabolite Analysis

Extraction and quantification of primary metabolites were performed in 6 repeats using gas chromatography–mass spectrometry (GC-MS) essentially as described by Lisec et al. [56]. Briefly, peels were ground in liquid nitrogen and lyophilized and the samples (90 mg) were extracted with 1 mL of a precooled mix containing methanol, chloroform, and MiliQ water (2.5:1:1 *v*/*v*, respectively) supplemented with ribitol as the internal standard (4.5 µg/mL) and vortexed thoroughly. Following incubation for 10 min at 25 °C on an orbital shaker, samples were sonicated for 10 min in an ultra-sonication bath at room temperature and centrifuged at high speed (10 min, 16,000× *g*). The supernatant was collected; 300 µL of MiliQ water and 300 µL of chloroform were added, vortexed for 10 s, and centrifuged for 5 min at high speed; and the upper phase was collected, aliquoted, lyophilized, and subjected to derivatization.

Derivatization was performed by adding 40 μL methoxyamine hydrochloride (20 mg/mL in pyridine) to the dry sample and incubating for 2 h at 37 °C on a shaker platform. Samples were added to 70 μL N-methyl-N-(trimethylsilyl)trifluoroacetamide (MSTFA) and 7 μL of alkane mix and incubated with constant shaking for 30 min at 37 °C. Samples were subjected to gas chromatography–mass spectrometry (GC-MS) analysis (Agilent Ltd., Santa Clara, CA, USA) as described [56,57]. Separation was carried out on a Thermo Scientific DSQ II GC/MS using a FactorFour capillary VF-5ms column (Agilent Ltd., Santa Clara, CA, USA). The acquired chromatograms and mass spectra were evaluated using Xcalibur (version 2.0.7) software and metabolites were identified and annotated using the Mass Spectral and Retention Time Index libraries available from the Max-Planck Institute for Plant Physiology, Golm, Germany (http://csbdb.mpimp-golm.mpg.de/csbdb/gmd/msri/gmd_msri.html accessed on 11 May 2021. PCA, ANOVA, Student’s *t*-tests, and hierarchical clustering analysis were performed using Metaboanalyst 4.0 [58]. The relative level of metabolites was calculated by normalizing the intensity of the peak of each metabolite to the ribitol standard.

### 4.8. Bacterial Growth Assay

The assay was performed essentially as described by Patton et al. [59]. We used *Escherichia coli* and *staphylococcus aureus* (obtained from Dr. Vardit Makover, microbiology lab at Ben Gurion University) as a representative model for Gram-negative and Gram-positive bacteria, respectively. One colony of each strain of bacteria was suspended in 10 mL of Luria broth (LB) medium and grown at 37 °C overnight. Later, the culture was diluted and transferred to 25% LB and grown at 37 °C until the optical density reached up to 0.03–0.05 (OD_595_; Epoch, Biotek, Winooski, VT, USA). A 150 µL aliquot of the culture was incubated with 50 µL of PBS, ampicillin (final concentration 50 µg/mL), or kanamycin (final concentration of 100 µg/mL), or with 50 µL of substances released from peels and cloves of garlic (4 replicates per treatment) in a flat-bottom 96-well microtiter plate. Plates were incubated in the dark using a spectrophotometer (Synergy 4, Biotek, Winooski, VT, USA), and reads (OD_595_) were taken for 12 h in 30-min intervals. The average OD for each blank replicate at a given time point was subtracted from the OD of each replicate treatment at the corresponding time points and standard errors were calculated for each treatment at every time point.

### 4.9. Fungal Spore Germination Assay

Microconidia of the pathogenic fungi *Fusarium oxysporum* f.sp. *melonis* (obtained from Dr. Omer Frenkel, Volcani Center, Israel) were derived from two-week-old mycelium cultivated on potato dextrose agar plates and diluted in double distilled water to a level of 10^4^ microconidia per mL. A total of 100 mg of outer peels, inner peels, and fresh cloves of *Allium sativum* were extracted with 1 mL of PBS, homogenates were centrifuged at maximum speed, and the sup was collected and analyzed for microconidia germination assay as follows. A total of 7 μL of microconidia suspension was mixed with 21 μL of potato dextrose broth solution (2.5%) and 42 μL of garlic extracts, vortexed, and co-cultivated in 96-well plates in a dark, moist chamber at 25 °C. Conidial germination was monitored after 16 h under a light microscope (Leica DFC7000 T, Europe) and the percentage of germinated spores and the hyphae growth were recorded (analyzed by counting 20 random fields under the microscope).

### 4.10. Statistical Analysis

Statistical analyses were performed by one-way ANOVA calculator with Tukey HSD (https://www.socscistatistics.com/tests/anova/default2.aspx; accessed on 13 November 2018).

## 5. Conclusions

The dead, papery-like outer garlic peel and the inner peel are highly valuable, and besides conveying a physical protective role, they appear to provide a rich maternal supply to ensure the success and survival of vegetative offspring in the habitat. Thus, a general theme in plant reproduction biology is to harness dead organs enclosing either the embryo (e.g., seed coats, pericarps, glumes) or the asexual reproductive body (e.g., outer garlic peel) to carry out multiple functions, including providing physical protection, dispersal means, and a refined arsenal of substances for enhancing growth and development; combating potential pathogens; and conferring tolerance to abiotic stresses. We are aware that due to their richness in bioactive substances, garlic peels and onion peels, which are often regarded as waste, might have the potential for utilization as a by-product [21,22,60,61,62].

## Figures and Tables

**Figure 1 ijms-23-02126-f001:**
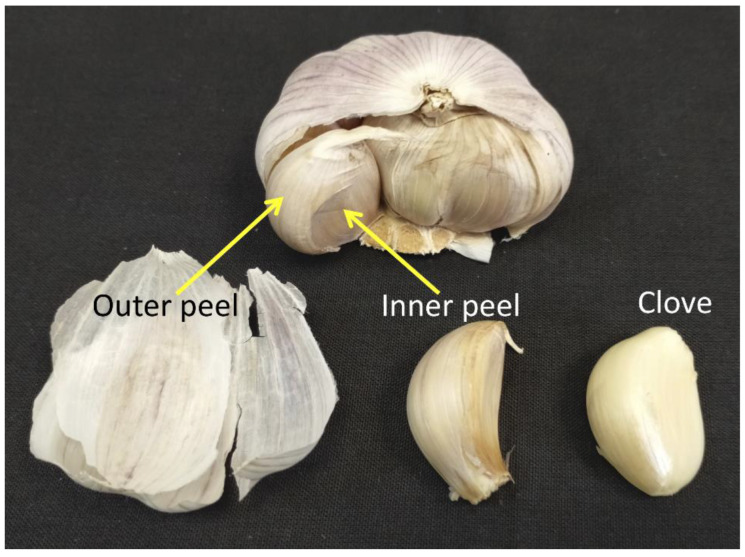
Garlic bulb. The outer and inner peels and the clove are indicated.

**Figure 2 ijms-23-02126-f002:**
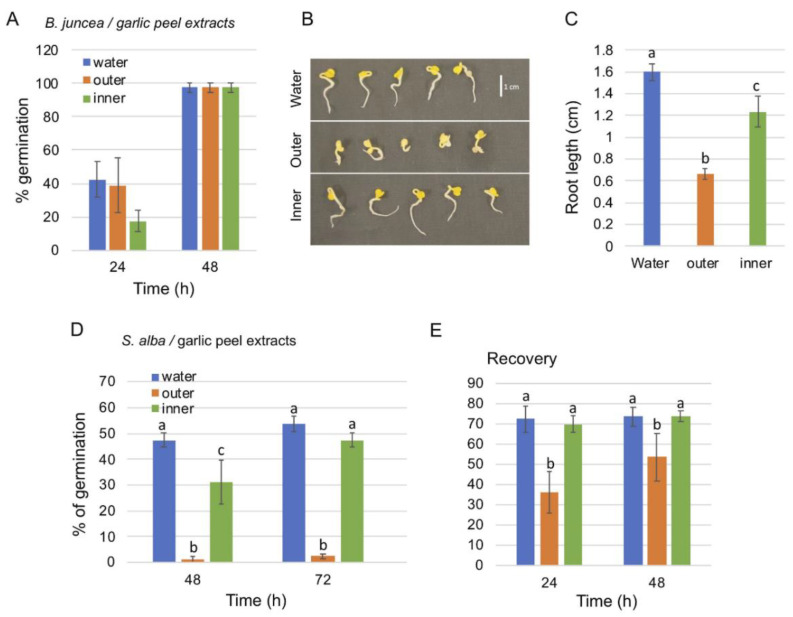
Effect of garlic peel extracts on seed germination of *Brassica juncea* and *Sinapis alba.* (**A**) Garlic peel extracts do not affect the final germination of *B. juncea* compared to water. (**B**) Post-germination growth of *B. juncea* is inhibited by outer peel extract. (**C**) Root length of germinating seeds after 72 h. (**D**) Outer peel extract strongly inhibits the germination of *S. alba* seeds. (**E**) Recovery of *S. alba* seed germination. Vertical bars represent the standard deviation. Statistical significance was performed by a One-Way ANOVA Calculator, plus Tukey HSD (Social Science Statistics). Different letters indicate statistically significant differences between treatments (*p* < 0.05).

**Figure 3 ijms-23-02126-f003:**
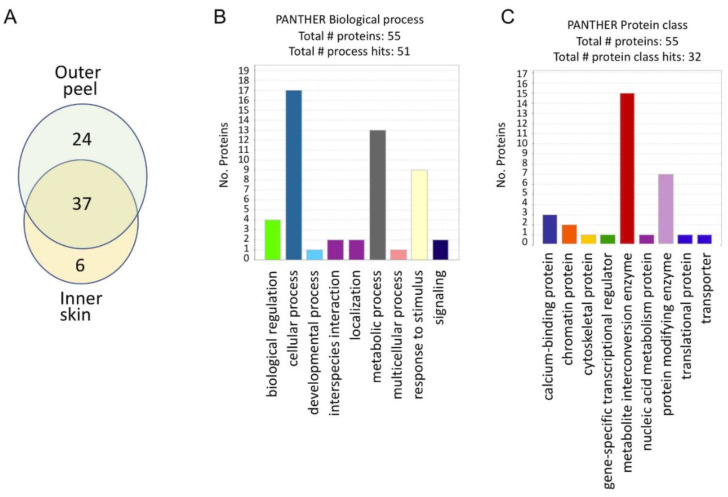
Proteome analysis of proteins extracted from outer and inner garlic peels. (**A**) Venn diagram showing the distribution of proteins between the outer and inner peels. (**B**) Categorization for the biological process of proteins recovered from garlic peels. (**C**) Categorization for protein class highlighting the high presentation of metabolite and protein-modifying enzymes. Categorization was performed with PANTHER v.16 [23].

**Figure 4 ijms-23-02126-f004:**
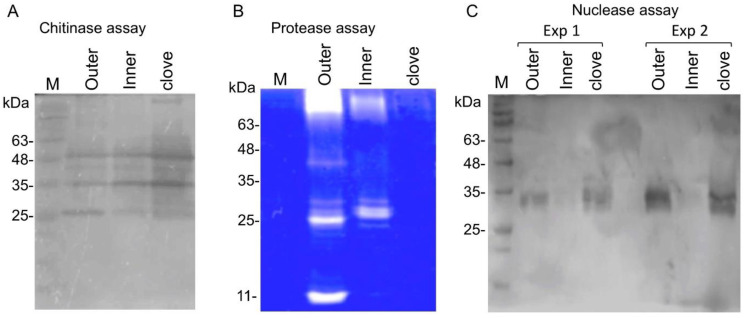
Hydrolytic enzyme activity in garlic peels and cloves. Proteins extracted from the garlic clove and the outer and inner peels were subjected to in-gel chitinase assay (**A**), in-gel protease assay (**B**), and in-gel nuclease assay (**C**). Note that two experimental repeats (Exp1 and Exp2) are shown for in-gel nuclease assay. M indicates protein molecular weight markers.

**Figure 5 ijms-23-02126-f005:**
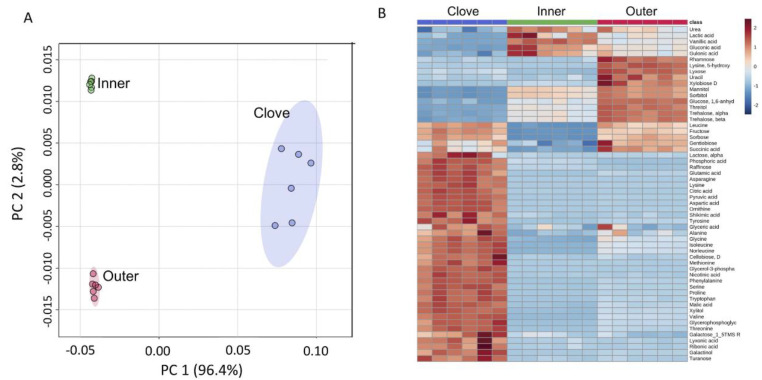
Primary metabolite analysis of garlic clove and the outer and inner peels. (**A**) PCA score plot comparing the 56 primary metabolites between the garlic clove and outer and inner peels. (**B**) Hierarchical clustering of metabolites recovered from clove and outer and inner peels. Metabolite levels are color-coded with brown and blue representing up and down accumulation, respectively. Each treatment (6 repeats) is represented by six columns.

**Figure 6 ijms-23-02126-f006:**
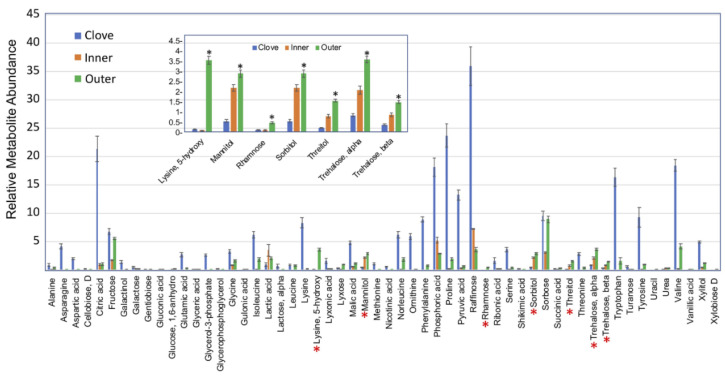
Relative metabolite abundance in garlic clove and inner and outer peels. Metabolites whose relative content is high in the outer peel compared to other organs are indicated by red asterisks and shown in the inset figure. Asterisks in the inset indicate statistically significant differences between the outer peel and other organs (*p* < 0.05). Statistical analysis was performed by One-Way ANOVA Calculator, plus Tukey HSD (Social Science Statistic).

**Figure 7 ijms-23-02126-f007:**
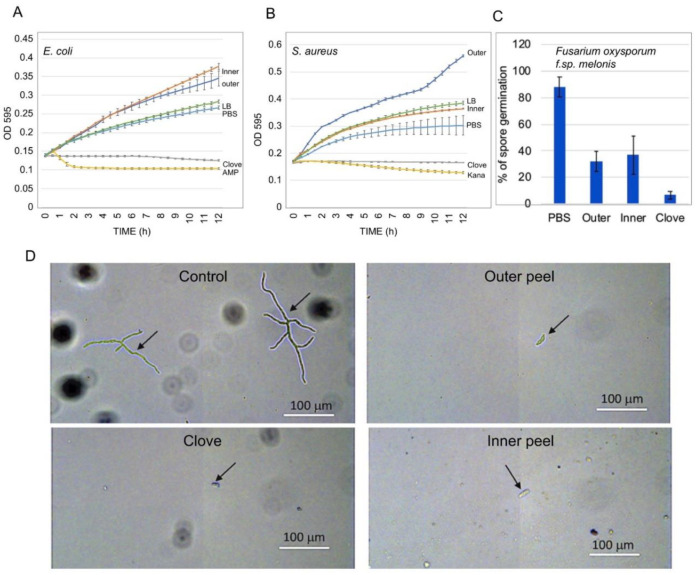
Garlic peel extracts contain substances that promote bacterial growth. *Escherichia coli* (**A**) and *Staphylococcus aureus* (**B**) were grown in a flat-bottom 96-well microtiter plate in the presence of PBS (control), or in the presence of garlic clove extract and outer and inner peel extracts. Ampicillin (Amp) and kanamycin (Kana) were used as antibiotic references. Bacterial growth (OD_595_) was measured at 0.5 h intervals over the course of 12 h. Each treatment was performed in triplicate and vertical bars represent the standard deviation. The effect of garlic clove and peel extracts on *Fusarium oxysporum f.sp. melonis* microconidia germination (**C**) and hyphae growth (**D**) is shown. Microconidia (asexual spores) were mixed with potato dextrose broth (PDB) alone (control) or with PDB supplemented with extracts derived from the outer and inner peels or from garlic cloves. Mixtures were placed on depression slides and incubated in a moist chamber for 16 h at 25 °C in the dark and inspected under a light microscope. Germinating spores and growing hyphae are indicated by arrows.

**Table 1 ijms-23-02126-t001:** List of proteins related to response to stimulus and to protein-modifying enzymes recovered from outer and inner garlic peels.

Biological Process/Response to Stimulus			
ID	Protein	Inner	Outer
A0A0Q3KZX7	CHITINASE	P	P
A0A0Q3RUL2	CHITINASE	P	P
A0A2K2D641	PEROXIDASE 64	A	P
D7NLB8	PEROXIDASE 1	P	P
I1H6 × 1	SHOCK COGNATE 70 KDA PROTEIN	P	P
I1HEK5	CALCIUM-BINDING PROTEIN CML9	A	P
I1HIQ7	HEAT SHOCK COGNATE 71 KDA PROTEIN	P	P
I1HWC7	HEAT SHOCK 70 KDA PROTEIN BIP1	A	P
I1IAA5	PEROXIREDOXIN-4	A	P
Prtein class/protein modifying enzyme			
**ID**	**Protein name**		
A0A0Q3INI3	SERINE PROTEASE	A	P
I1GNN5	CYSTEINE PROTEASE	A	P
I1HWA5	SERINE PROTEASE	A	P
I1I5B9	SERINE PROTEASE	A	P
I1IQ27	CYSTEINE PROTEASE	P	P
I1ITB7	SERINE PROTEASE	A	P
I1J2L7	CYSTEINE PROTEASE	P	P
A0A0Q3H6A6	SERINE PROTEASE	A	P

## Data Availability

The data that support the findings of this study are available in the main text and in Appendix A.

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
