# Peer review of "What Worth the Garlic Peel"

_ijms, 2022, doi:10.3390/ijms23042126_

Round 1

Reviewer 1 Report

In this paper, the scientists used the garlic (Allium sativum) bulb as a model to investigate the features of dead organs enclosing the asexual reproductive body. The biochemical and biological features of garlic peels were examined, as well as the functional similarities between garlic peels and DOEEs. This is a well-written and informative manuscript about such unknown garlic peels including bioassays, proteomics, and metabolomics. There is no doubt about the study's importance and the protocols designed to reach the goals. My minor concerns are about the in-gel nuclease assay (I recommend performing new electrophoresis, if possible), and the order of hydrolytic enzymes activity in garlic peels and cloves in M&M should follow its appearance in Results and Discussion. Final of Page 1 (44-47) and 11 (338-343) there is a mistake in size letters. Figure 6 should be bigger for better visualization of relative metabolites abundance and comparison. The conclusion could be remade, especially the final sentence, which in my opinion could be considered in the discussion section, and so the authors consider future prospects in the research area about peels. Some reference updated is required as well.

Author Response

Thanks for your comments.

Reviewer 1

In this paper, the scientists used the garlic (Allium sativum) bulb as a model to investigate the features of dead organs enclosing the asexual reproductive body. The biochemical and biological features of garlic peels were examined, as well as the functional similarities between garlic peels and DOEEs. This is a well-written and informative manuscript about such unknown garlic peels including bioassays, proteomics, and metabolomics. There is no doubt about the study's importance and the protocols designed to reach the goals.

  1. My minor concerns are about the in-gel nuclease assay (I recommend performing new electrophoresis, if possible), and the order of hydrolytic enzymes activity in garlic peels and cloves in M&M should follow its appearance in Results and Discussion.

Response: We are confident that the image demonstrating the in-gel nuclease assay faithfully conveys the core results.  We changed the order of hydrolytic enzymes methodology in M&M to fit with the Results and Discussion.

  1. Final of Page 1 (44-47) and 11 (338-343) there is a mistake in size letters.

Response: Corrected

  1. Figure 6 should be bigger for better visualization of relative metabolites abundance and comparison.                                                                         Response: Technically, in its present orientation, it is difficult to increase the font size in Fig. 6, though changing orientation would do that.
  2. The conclusion could be remade, especially the final sentence, which in my opinion could be considered in the discussion section, and so the authors consider future prospects in the research area about peels. Some reference updated is required as well.

Response: Since we did not address the potential use of garlic peels as a by-product in the present work, we do not think it is appropriate to discuss in length this important aspect. We included two updated references (published at 2021) for this matter and added another recent one (ref. 62, year 2022).

Reviewer 2 Report

The paper is clear and well written.

I only sugges to revise it for typing errors

Author Response

Thanks!

Comments and Suggestions for Authors

The paper is clear and well written.

I only suggest to revise it for typing errors.

Response: Thanks. We passed through the manuscript and corrected typing errors.

Reviewer 3 Report

Dear Editor,

Enclose please fint the review of the Manuscript Id: ijms-1572024

Reviewer'r report on: What worth the garlic peel

I revised this MS and here are my comments:

The aim of the study is the properties of the dead organs surrounding the asexual reproductive body - the garlic bulb (Allium sativum L.). The subject is appropriate for the journal. The results are new, and the material and methods are well presented. Several assays are performed in the study: Germination assay, proteomic analysis, in-gel nuclease assay, in-gel hitinase assay, in-gel protease assay, metabolite analysis, bacterial growth assay, and fungal spore germination assay.

The results are well presented and the discussion is comprehensive enough.

The authors concluded that the outer and inner shells of garlic are very valuable, providing not only a physical protective function but also a rich supply of material that ensures the success and survival of vegetative progeny in the habitat. They also concluded that garlic and onion peels, which are often considered waste, have the potential to be utilized as a byproduct.

Author Response

Thanks! no further response.